# When are smooth-ReLUs ReLU-like?

## Abstract

ReLU is one of the most popular activations in deep learning, especially thanks to its stabilizing effect on training. However, because it is non-differentiable at the origin, it complicates the use of analysis methods that examine derivatives, such as the Neural Tangent Kernel (NTK). Many smooth relaxations try to retain the practical benefits of ReLU while increasing network regularity. Although their success has ranged widely, some notable architectures (e.g., the BERT family) do utilize them. We present a theoretical characterization of smooth-ReLUs within fully-connected feed-forward neural networks. In addition to the well-known SWISH and GeLU, we introduce GumbelLU, AlgebraicLU, and GudermanLU, as new relaxations. All these activations can be characterized by a positive temperature parameter which we can lower to continuously improve the approximation. By studying the interplay of initialization schemes with temperature, we confirm that when these relaxations converge to ReLU, the statistical properties of the corresponding neural networks at initialization also converge to those of ReLU networks. Moreover, we derive temperature-dependent critical initialization schemes with which networks based on these activations exhibit stable ReLU-like behavior at any temperature. Finally, we empirically study both classes of networks on MNIST and CIFAR-10 in the full-batch training regime. We observe faster training dynamics of smooth-ReLU networks with our proposed initialization instead of the standard one. While all networks exhibit very similar train loss trajectories at criticality, smooth-ReLU networks feature differentiable NTKs throughout training, whereas ReLU networks exhibit stochastic NTK fluctuations. Our results clarify how smooth-ReLU relaxations reproduce the practical benefits of ReLU in everywhere-smooth neural networks.

## 1 Introduction

In recent decades, deep learning has shown tremendous success in e.g. computer vision, natural language processing, and drug discovery (LeCun et al., 2015). For instance, EfficientNet (Tan and Le, 2019) achieved state-of-the-art performance in image classification on CIFAR100 (Krizhevsky et al., 2009) and ImageNet (Deng et al., 2009). In natural language processing, GPT (Radford et al., 2018) and its newer versions (Radford et al., 2019; Brown et al., 2020) were capable of producing human-like responses to various reading comprehension tasks. As the name suggests, deep learning neural architectures are composed of many layers sequentially applied. The most basic architecture, the fully connected feedforward network (FFN), consists of an alternating sequence of linear layers and non-linear layers called activations. One of the most popular activations is ReLU (Jarrett et al., 2009; Nair and Hinton, 2010), given its computational simplicity, gradient stability, and expressive power (Nair and Hinton, 2010; Raghu et al., 2017). Beyond ReLU, EfficientNet and others use SWISH (Ramachandran et al., 2017; Elfwing et al., 2018; Alcaide, 2018; Chieng et al., 2018; Howard et al., 2019). GPT-2 and ALBERT (Lan et al., 2020) employ the GeLU activation (Hendrycks and Gimpel, 2016). Both are smooth approximations to ReLU, with the advantage of the existence of higher-order derivatives to study their properties (Hanin and Nica, 2020b; Li et al., 2021). Activation smoothness is required to ensure smoothness of the network's input-output mapping; this is in turn necessary for certain applications (e.g., physics-informed neural networks and neural network methods to solving differential

equations require the network output to be a smooth function of its input, (Raissi et al., 2019)), certain architectures (e.g., the existence of neural ODEs is typically proven assuming smooth activations (Chen et al., 2018)), and for certain theoretical analysis techniques (e.g., differential Neural Tangent Kernel (NTK) approaches Roberts et al. (2022)).

Our focus is to understand the properties of smooth ReLU relaxations, that we call smooth-ReLUs, of the type $\sigma_T(z) = za(z/T)$, with $T$ a positive *temperature* parameter and $a(\cdot)$ any sigmoid function ranging from 0 to 1. As the temperature is lowered, the smooth function $a(z/T)$ converges to the Heaviside function $H(z)$ pointwise. Different choices of $a(\cdot)$ correspond to different smooth approximations to ReLU. The well known SWISH and GeLU can be represented this way, and we introduce new activations with this property. Inspired by recent works based on renormalization group and field theory (Roberts et al., 2022), we study the stability of information propagation through the layers of FFNs with smooth-ReLU variants. In particular, we study statistical properties over random network initializations, compare initialization and training dynamics of various smooth-ReLUs, and analyze to what extent they echo ReLU. Initializing weights and biases by sampling a standard normal distribution results in the exponential explosion of the variance of the representations. Weight initializations whose variance is proportional to the inverse of the width can train deeper linear networks, more reliably and rapidly, by keeping the variance of representations and gradients constant with depth. However, for non-linear activations, an additional proportionality factor, that is activation dependent, is necessary to obtain depth stability (He et al., 2015; Roberts et al., 2022). Based on this observation, we quantify how similar different smooth-ReLU functions are to ReLU by comparing the variance of network representations at initialization. We also compare the behaviour of training loss during full-batch training. To gain further insight into training dynamics, we also use the NTK theory (Jacot et al., 2018). In recent years, kernel methods have found wide use to theoretically understand neural networks' performance (Allen-Zhu et al., 2019; Zou et al., 2020; Liu et al., 2022; 2020; Yang, 2020). A flurry of works have extended the NTK analysis to all kinds of neural networks, with Arora et al. (2019) extending NTK to convolutional networks, and more recently Feng and Kolter (2020) investigating the NTK analysis for large-depth limits in Deep Equilibrium Models Bai et al. (2019).

Our work aims to improve our understanding of the expressivity and training of fully-connected deep smooth-ReLU networks and devise simple prescriptions to improve them. Specifically, our contributions are as follows:

- Derive conditions of existence of stable initialization schemes for smooth-ReLUs at any temperature, and provide code for computing the necessary hyper-parameters.
- Introduce GumbelLU, AlgebraicLU and GudermanLU. We note that, in certain aspects of the analysis, GudermanLU is closer to ReLU than the other smooth-ReLUs analysed, including SWISH and GeLU.
- Demonstrate experimentally that representation variance and NTK of smooth-ReLUs resemble those of ReLU under the stable initialization scheme we provide, for any temperature. Also, the same applies for very low temperatures with standard initialization.
- Show that smooth-ReLUs typically have faster training dynamics with our proposed initialization, and stable and continuous NTK updates. ReLU's NTK shows stochasticity instead. We observe that the smooth-ReLU NTK updates under our stable initialization are similar for different temperatures.

## 2 PRELIMINARIES

### 2.1 INITIALIZATION COVARIANCE

To better understand the trainability of a FFN, we pay attention to how the magnitude of the input propagates through the layers. Let's denote the layer preactivation as,

$$z_{i;\alpha}^{(l)} = \sum_{j}^{n} W_{ij}^{(l)} \sigma(z_{j;\alpha}^{l-1}) + b_i^{(l)}, \tag{1}$$

where $i, j$ denote neuron indices, $\alpha_k$ data indices, and $l$ and $n$ denote depth and width respectively. The activation function is denoted by $\sigma$. We assume the network weights and biases are drawn from normal distributions with means zero and variances $\mathbb{E}\left[b_i^{(l)} b_j^{(l)}\right] = \delta_{ij} C_b^{(l)}$ and $\mathbb{E}\left[W_{i_1 j_1}^{(l)} W_{i_2 j_2}^{(l)}\right] = \delta_{i_1 i_2} \delta_{j_1 j_2} C_W^{(l)}/n_{l-1}$. For simplicity, we assume all layers to have equal width $n_l = n$, possibly different from the input size $n_0$, and the same initialization hyper-parameters, i.e. $C_b^{(l)} = C_b$ and $C_W^{(l)} = C_W$.

We are interested in the magnitude of the second order correlation metric defined as

$$K_{\alpha_1 \alpha_2}^{(l)} = \left\langle \frac{1}{n} \sum_{j=1}^{n} z_{j;\alpha_1}^{(l)} z_{j;\alpha_2}^{(l)} \right\rangle, \tag{2}$$

where $\langle \cdot \rangle$ denotes ensemble averaging with respect to the initializations of the weights of the architecture, and $\alpha_i$ denote different input data samples for different $i$. In simple terms, one has to sample the network parameters from the normal distribution multiple times and then compute population statistics. Intuitively, this metric measures magnitude change per layer of the layer output, or representation. Given the progressive transformations through layers, the following recursion is satisfied

$$K_{\alpha_1 \alpha_2}^{(l+1)} = C_W \left\langle \sigma(z_{\alpha_1}) \sigma(z_{\alpha_2}) \right\rangle_{K = K_{\alpha_1 \alpha_2}^{(l)}} + C_b \tag{3}$$

as shown in Theorem 2.2 in (Hanin, 2022). The subscript $K$ indicates integration with a Gaussian distribution of zero mean and $K$ variance, see App. A. For clarity, we omit input subscript, unless required. When this metric remains constant in magnitude with depth, we call the architecture stable, and the initialization as critical.

As an illustration, we take into consideration linear networks, where the activation is the identity function. In these networks the metric behaves like a scalar dynamic system. The resulting representation variance is $K_{\alpha_1 \alpha_1}^{(l+1)} = C_W K_{\alpha_1 \alpha_1}^{(l)} + C_b$. If $C_W > 1$, the metric grows exponentially with depth. Instead, if $C_W < 1$, the metric vanishes exponentially to zero with depth, eliminating any data-dependence in the outputs. Finally, if $C_W = 1$ (Glorot and Bengio, 2010), it prevents both, the exponential explosion and vanishing of the variance. In other words, it is a critical initialization. For the same reason, in a non-linear neural network, if the metric increases or decreases exponentially, the variance either diverges to infinity or converges to zero. Both outcomes are undesirable as they lead to exploding or vanishing representations and gradients, and make training very difficult.

In addition, one may be concerned with higher order correlations of the activation and how they grow with depth. For instance, in the infinite-depth-and-width limit for fully connected networks with ReLU activation, the statistics is characterized by the log-Gaussian distribution (Hanin and Nica, 2020a). Here we focus on the $l/n \ll 1$ regime with $l \gg 1$, where we can safely assume that the quadratic moment dominates the distribution (Roberts et al., 2022). In App. H we compute higher moments and confirm this assumption experimentally.

## 2.2 Neural Tangent Kernels

The Neural Tangent Kernel (NTK) theory is widely used to study the dynamics of infinitely-wide deep neural networks under gradient descent (Jacot et al., 2018; Seleznova and Kutyniok, 2020). Here we focus on the finite width version defined as,

$$\Theta_{i,\alpha_1;j,\alpha_2} = \sum_{\mu,\nu} \partial_{\phi_\mu} z_{i;\alpha_1}^{(l)} \partial_{\phi_\nu} z_{j;\alpha_2}^{(l)} \tag{4}$$

where $\phi$ represents the parameters of the network, and we sum over all the intermediate layer parameters. For the samples collected we compute the ensemble average,

$$\tilde{\Theta} = \left\langle \frac{1}{n} \sum_{i,j,\alpha_1,\alpha_2} |\Theta_{i,\alpha_1;j,\alpha_2}| \right\rangle \tag{5}$$

Critical initialization dynamics of reasonably large neural networks can be well-approximated by results corresponding to the NTK regime (Roberts et al., 2022). Although the true optimal initialization schemes will differ from those derived based on the infinite-width approximation, the corrections will generally be of order $\mathcal{O}(l/n)$. For networks with aspect ratios $l/n \ll 1$, the critical initialization schemes derived in this work will be very good approximations, as demonstrated in the following sections. Introducing a new index $t$ to indicate training iteration, we define the NTK update as

$$\Delta \tilde{\Theta}_t = \tilde{\Theta}_t - \tilde{\Theta}_{t-1} \tag{6}$$

In general, during training, there is an initial phase of rapid decrease in the loss function followed by a longer phase of very small updates. Based on this observation, and since $\Delta \tilde{\Theta}_t$ is related to the parameters update, we expect it to first increase in the initial epochs and proceed to decrease in magnitude as learning improves. As such, analysing this 'observable' gives another useful indication of whether training converges, in addition to the loss function curve. For a very small learning rate, we have $d\tilde{\Theta}/dt \approx \Delta \tilde{\Theta}_t$, and we expect it to be continuous for smooth functions, but not so for ReLU which is discontinuous at zero. As our main focus is the study of the initilization and training of fully connected networks with finite width and depth, our NTK is dynamical and dependent on the training iteration. In other words, it does not only apply to initialization statistics as can be seen from Setion 4.3.2. In addition, the differential of the NTK and the Fisher Information Matrix (FIM) (Fisher, 1920) can be useful in developing new training methods (Amari, 1998). They would further display the finite size effects of width and depth but are beyond the scope of this first work.

## 3  THEORY FOR SMOOTH-RELU CRITICAL INITIALIZATION

As mentioned, we consider the initialization to be critical when it allows the network to keep the variance constant with depth, i.e. $K_{\alpha_1 \alpha_1}^{(l+1)} = K_{\alpha_1 \alpha_1}^{(l)} = K^*$, $K^*$ being the fixed-point value (Glorot and Bengio, 2010; He et al., 2015; Roberts et al., 2022). Here, we are interested in keeping the diagonal components of $K_{\alpha_1 \alpha_2}^{(l)}$ constant. In App. E and App. C, we prove the following Lemma on the conditions that have to be satisfied for a network with this type of activations to admit critical Gaussian initializations:

**Lemma 1** (Criticality Condition). *A FFN network with activation $\sigma(z) = za(z)$ admits a critical initialization scheme if and only if the following conditions are satisfied*

- $2\langle za(z)a'(z)\rangle_{K^*} = -\langle z^2 a(z)a''(z)\rangle_{K^*}$
- $z^3 a(z)^2$ *and* $a(z)(a(z) + za'(z))$ *are sub-Gaussian* $(\lim_{z\to\pm\infty} f(z)e^{-z^2/2K} = 0)$

Notice that even if the criticality exists, it might have an undesirable $K^* = 0$, as it is the case for $a(z) = \arctan(z)$ function, see App. E. Therefore:

**Lemma 2** (Zero Criticality). *For neural networks with activations of the form $za(z/T)$, the condition that $a(z/T) \to H(z)$ as $T \to 0$ is not sufficient to ensure the existence of a critical initialization scheme with a non-zero fixed point $K^* \neq 0$.*

We prove that the critical covariance and hyper-parameters can be uniquely determined for all temperatures, if they are known for a specific temperature, which we chose to be $T = 1$ for reference:

**Lemma 3** (Temperature Criticality Smooth ReLU). *If the activation $\sigma(z) = za(z)$ is critical at $K^*$ for $(C_W^*, C_B^*)$, then the activation $\sigma_T(z) = za(z/T)$ is critical at $K_T^* = K^*T^2$ for $(C_{W,T}^*, C_{B,T}^*) = (C_W^*, T^2 C_B^*)$.*

A simple consequence is that both the critical representation covariance $K^*$ and bias variance $C_B^*$ tend to zero as we decrease the temperature:

**Corollary 1.** $\lim_{T\to 0} K_T^* = \lim_{T\to 0} C_{B,T}^* = 0$.

This is an interesting outcome if we keep in mind that in ReLU networks any value of $K$ serves as a fixed point when critically initialized with $C_W = 2$ and $C_b = 0$ (He et al., 2015),

known as He initialization. For smooth-ReLU networks the critical variance tends to zero as the temperature approaches zero and the activation becomes closer in shape to ReLU. However, as we show in the next section in Fig. 1, the scaled layer update, $\Delta K/K^*$, shows the same dependence at both $T = 0.1$ and $T = 1$. In other words, the scaled quantities in this figure have no temperature dependence. Since $K^*$ is proportional to $T^2$, so is the layer update proportional to $T^2$. As such, the $\Delta K$ tends to zero at low temperatures, effectively making every $K$ behave like a fixed point at $T = 0$.

In Table 1 we show several differentiable relaxations of ReLU, in addition to the well known SWISH, GeLU and Mish (Misra, 2019), together with their critical variances and hyper-parameter initializations. These values were obtained for infinitesimal variations as shown in App. A. In App. F we compute analytically the critical point for GeLU. We introduce GumbelLU, AlgebraicLU, and GudermanLU, by simply observing that there are many functions that can serve as smooth relaxations of the Heaviside function with a temperature parameter. Strictly speaking, these activation functions are different from ReLU, judged for example by $C_W^*$, which for ReLU has an exact value of $C_W^* = 2$, while it is very close to 2 but never exactly so for its relaxations. ELU (Clevert et al., 2015), SoftPlus (Dugas et al., 2000), and the exponential activations cannot be tuned to criticality, since they do not satisfy the first condition of Lemma 1, and thus, are not present in the table. However, the formulas we use to calculate $C_W^*$ and $C_b^*$ are based on first-order perturbation theory, and the inclusion of higher orders could bring $C_W^* = 2$. As we show in this work, the difference in $C_W^*$ is rather small for practical purposes. The last column comments on the stability of the critical point, and semi-stable refers to the fixed point being stable only for perturbations to the left or right, but not both, see App. B.

| Activation | $a(u)$ | $K^*$ | $C_b^*$ | $C_W^*$ | Stability |
|---|---|---|---|---|---|
| SWISH | $(1 + \exp(-u))^{-1}$ | 14.320 | 0.555 | 1.988 | Semi-stable |
| GeLU | $\frac{1}{2}\left(1 + \text{Erf}(\frac{u}{\sqrt{2}})\right)$ | $\frac{1}{2}\left(3 + \sqrt{17}\right)$ | 0.173 | 1.983 | Semi-stable |
| GumbelLU | $\exp\left(-\exp\left(-u\right)\right)$ | 21.123 | 0.606 | 1.988 | Semi-stable |
| AlgebraicLU | $\frac{1}{2}\left(\frac{u}{\sqrt{(u)^2+1}} + 1\right)$ | 20.210 | 0.334 | 2.005 | Unstable |
| GudermanLU | $\frac{1}{2} + \frac{2}{\pi}\tan^{-1}\left(\tanh\left(u\right)\right)$ | 3.154 | 0.103 | 1.990 | Semi-stable |
| Mish | $\tanh(\ln(1 + \exp(u)))$ | 1.670 | 0.094 | 2.013 | Unstable |
| ReLU (He) | Heaviside(u) | — | 0 | 2 | Stable |

Table 1: Critical hyper-parameters for different smooth-ReLU activations. The more similar $C_b^*$ and $C_W^*$ are to their ReLU counterpart, the more ReLU-like the activation can be considered to be in the low temperature limit or when critically initialized, see App. J.

## 4 Numerical Results

### 4.1 Metric Layer Update

To better understand how the input magnitude changes with depth, we define the layer update of the metric as $\Delta K = K^{l+1} - K^l$ where $C_W^*$ and $C_b^*$ are tuned to criticality for a given temperature. The experimental results in this and the following subsection were computed from 20 ensembles of 1000 samples each. The input data was sampled from a standard normal distribution and kept the same across all architecture samples, that resulted in $K$ spanning values below $K^*$ up to $100K^*$. In Fig. 1 we compare the experimental layer update, denoted by points with one standard deviation uncertainty bands, and theoretical prediction in dashed lines, see App. A, for SWISH and GeLU, $T = 0.1, 1$, and depth $l = 10$. The horizontal axis denotes the magnitudes of the output of the last layer. As the figure confirms, the theoretical and experimental re-scaled update match, for all temperatures in the entire domain. As shown in the App. B, very near $K^*$ there is another numerically critical point, except for AlgebraicLU, and both points scale together with temperature. There is no qualitative difference between various temperatures and only at $T = 0$ these two fixed points are both equal to zero. As a reminder, for ReLU, once we set $C_W = 2, C_B = 0$,

any value of $K$ serves as a fixed point since there is no variance magnitude change between layers. For all the activations in Table 1, $\Delta K \propto K$ for $K \gg K^*$ or $K \ll K^*$, so the update remains exponential in nature, but the exponent is a rather small value. In practice, a deep, but not terribly deep, neural architecture, if properly initialized, should be both expressive and trainable. As an additional confirmation, we also compute the layer update for ReLU initialized at the critical parameters of the two other activations. It is remarkable how well it matches them throughout the domain, with the biggest discrepancies at small values of $K^*$, in the inset.

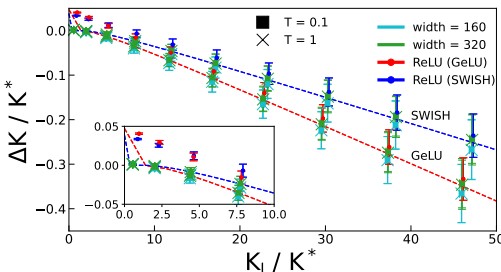

Figure 1: Layer update experimental results, for $T = (0.1, 1)$, critical initializations, for the SWISH and GeLU activation functions. The theoretical results in dashed lines match surprisingly well the experimental values. For GumbelLU see App. I.

To compare critical and He initializations in practice, in Fig. 2 we depict the layer update for He, at depth $l = 10$ and width $n = 320$, at the two temperatures from the previous figure. As a reminder, at $T = 0.1$ the activations look similar to ReLU, while at $T = 1$ they have their natural shape. At $T = 0.1$ we recover the stable critical behavior of ReLU for all three activations. The layer update is essentially zero within the uncertainty band, which can be further reduced by increasing sampling size. At $T = 1$ the two activations are very different from ReLU. For both temperatures, the layer update is negative, which indicates that the magnitude decreases with depth. In summary, ReLU networks initialized with smooth-ReLU critical hyper-parameters behave like them except at very low values of $K$. Also, smooth-ReLUs initialized with ReLU critical hyper-parameters behave like ReLU only at low temperatures. smooth-ReLUs initialized with their respective hyper-parameters are markedly different from ReLU with He initialization, but the magnitude of the layer update, $\Delta K$, is very small with respect to $K$. We have confirmed this empirically for GeLU and SWISH in Fig. 1 and Fig. 2.

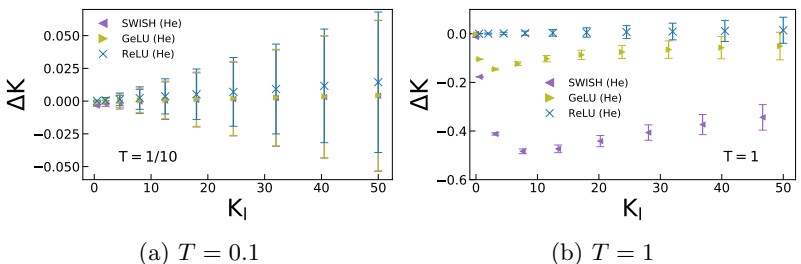

(a) $T = 0.1$                    (b) $T = 1$

Figure 2: Experimental results on the layer update for $T = 0.1$, He and critical initializations, at depth $L = 10$ and width $N = 320$. Figure (a) shows how the two relaxations converge to ReLU behavior at low temperatures for the He initialization. In figure (b) we plot them for $T = 1$, and notice sharp differences between each other and ReLU. Both cases confirm that at low temperatures ($T \leq 1$) architectures with continuous relaxations are both differentiable throughout the domain and behave like ReLU; see App. I. for GumbelLU.

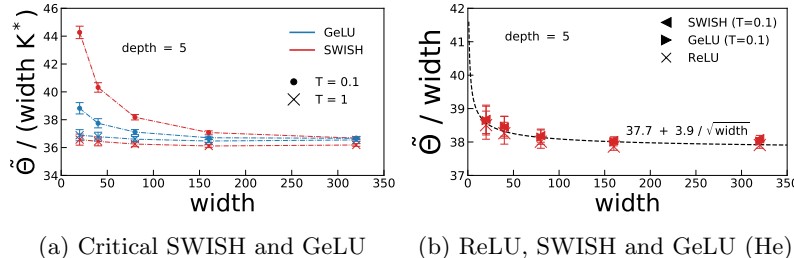

(a) Critical SWISH and GeLU    (b) ReLU, SWISH and GeLU (He)

Figure 3: Experimental results for the NTK at $T = 0.1$ and $l = 5$ as function of width. Figure (a) shows the SWISH and GeLU activations initialized at criticality, and figure (b) compares ReLU with SWISH and GELU with He initialization. Both cases confirm that at low temperatures architectures with continuous relaxations are differentiable throughout the domain and behave like ReLU when initialized in the same way.

## 4.2    NTK at Initialization

In sub-figure 3 (a) we plot ensemble averaged NTK for critical initializations of SWISH and GeLU, as a function of width, for a fixed depth $l = 5$. As it can be seen, $\tilde{\Theta}/(nK^*)$ is approximately a constant with some deviation for smaller width values, irrespective of temperature. Then, in sub-figure (b) we compare them at $T = 0.1$, to ReLU under He initialization. We fit $\tilde{\Theta}/n$ to confirm the constant behavior in the previous plot. We do notice a sub-leading $\sim n^{-1/2}$ which quickly disappears as width increases. It is remarkable that in both cases we observe a constant $\tilde{\Theta}/n$. Not only the magnitude of the output, $K$, but also the gradients with respect to the parameters, $\tilde{\Theta}$, are very similar between FNN with ReLU and the smooth approximations we study, for low temperatures. We have numerically confirmed the same behaviour to be true for all the functions listed in table 1

## 4.3    Training dynamics

The previous section confirmed that smooth-ReLUs are ReLU-like at initialization, if the network is tuned to criticality. Here we study their training properties. We focus on the CIFAR10 (Krizhevsky et al.) and MNIST (LeCun and Cortes, 2010) datasets. To remove mini-batch stochasticity and better resolve the dynamics, we train with full-batch gradient descent at fixed learning rate of $10^{-3}$. Due to computational memory limitations, we select the first 100 images for each category for training, and 10 samples per category for testing. To understand the impact of initialization, we compare the learning curves for He (critical for ReLU), and the SWISH critical initialization, for a wide range of temperatures. The trends we observe are the same across various seeds for the random number generator. For clarity of presentation, all the plots displayed in this section were obtained with seed equal to 1. We tested with various depths from 4 to 16 and for the presentation that follows we display results for $l = 8, 12$. The widths were set equal to the number of pixels of the input, 784 for MNIST and 1024 for CIFAR10. The cost function is the negative log-likelihood. Here we show the training curves for SWISH and ReLU on CIFAR10, and other activations can be found in App. J, where we also show results for the MNIST dataset.

### 4.3.1    Training Loss function

Fig. 4 shows the training curves on the CIFAR10 dataset, for SWISH and ReLU activations. When using He initialization, training worsens as the temperature is increased. In other words, as the activation becomes increasingly different from ReLU, training takes longer to converge. As the depth is increased, learning becomes harder for these high-temperature, He-initialized smooth-ReLU networks. This finding agrees qualitatively with what we observed in Fig. 2: at low temperatures SWISH behaves like ReLU (and training is stable) while at higher temperatures it becomes less stable (slower learning made worse by greater depths). When SWISH is critically initialized, all loss curves are closer to each other, for

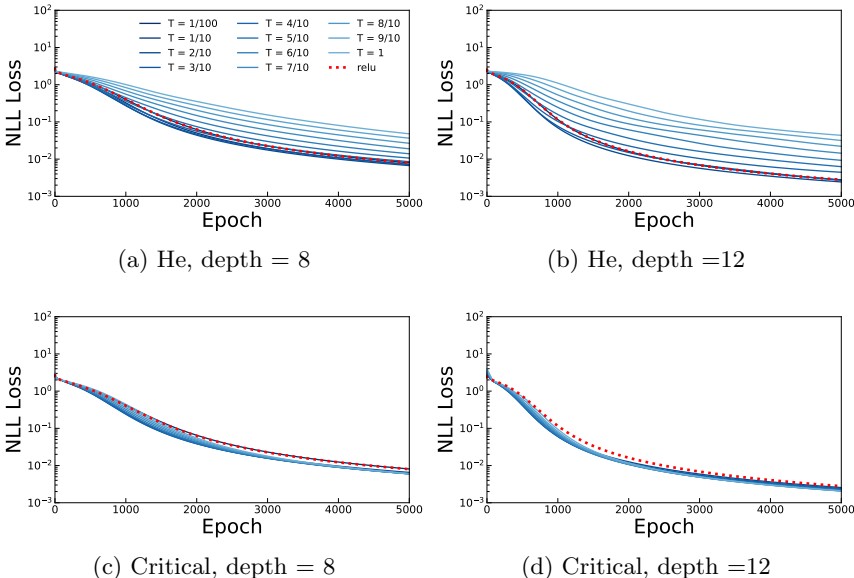

(a) He, depth = 8

(b) He, depth =12

(c) Critical, depth = 8

(d) Critical, depth =12

Figure 4: Loss function for Cifar10 training with SWISH activation, He and critical initialization.

all temperatures and for ReLU. In both initialization schemes we notice that the deeper networks have a steeper decrease of loss in the first 1000 epochs region. In addition, critical initialization leads to faster training. In Fig. 4 we compare the training curves in the first row versus the second row for the same temperatures. At T = 1, for instance, the critically tuned networks reach a loss of $10^{-1}$ approximately five times faster with respect to the He initialization case. In addition, we trained the small Transformer (Vaswani et al., 2017) on the WMT'14 English-German translation (Bojar et al., 2014) and the B0 EfficientNet (Tan and Le, 2019) on CIFAR100 with GudermanLU and SWISH for the He initialization, see App. K. The GudermanLU's loss curve stays closer to ReLU, as expected, given that both $C_W^*$ and $C_b^*$ are closer to ReLU's values. Also, lowering the temperature moves both training curves closer to ReLU.

### 4.3.2 Neural Tangent Kernel

The one step update of the empirical NTK, $\Delta\tilde{\Theta}_t$ during training is plotted in Fig. 5. Networks with ReLU activations have highly irregular NTK variations over time, such that the differentials are not well-defined. In contrast, networks with smooth activations have smoothly varying NTKs throughout training, and this is crucial for a theoretical understanding of all stages of training. Any training theory based on the NTK and its differentiability should be applicable to smooth-ReLU networks at every non-zero temperature. Understanding the convergence of these networks to ReLU networks would allows us to extend such understanding to ReLU networks as well. The $\Delta\tilde{\Theta}_t$ for smooth-ReLUs first increases and then proceeds to decrease, as the task is learned and convergence achieved. For He there is a clear temperature dependence and the NTK update gets larger with increasing $T$ and $l$, while for the critical initialization the various curves are closer to each other and less sensitive to depth. This result is a further confirmation of our observations throughout this work. Remarkably, very low temperature SWISH, $T = 0.01$, starts showing some level of stochasticity in the $\Delta\tilde{\Theta}_t$ as well. As the ReLU NTK shows no clear trend, in App. J Fig. 10, we apply a Savitzky-Golay filter (Savitzky and Golay, 1964; Virtanen et al., 2020) with a window of 80 epochs and a polynomial order of 3, to improve the clarity of the plot. We tested several epochs and polynomial orders for the window filtering with similar results. After using the window filter, in the initial faster training phase, the ReLU follows

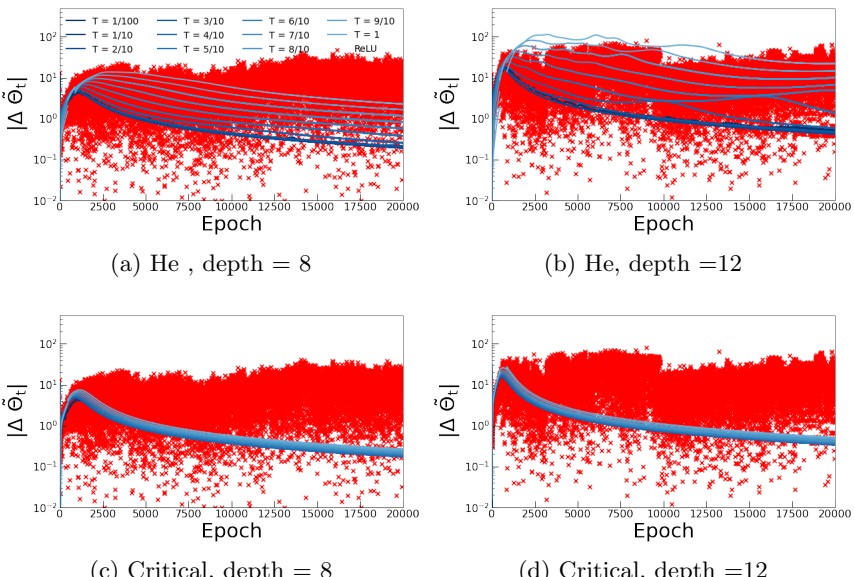

(a) He , depth = 8

(b) He, depth =12

(c) Critical, depth = 8

(d) Critical, depth =12

Figure 5: NTK update for Cifar10 training with SWISH activation, He and critical initialization.

closely the low temperature SWISH behavior, while on the later stages it seems to resemble stochastic noise.

## 5  Summary and Conclusion

In this work we study differentiable approximations to ReLU, in fully-connected feedforward neural networks and restrict ourselves to a temperature interpolation of the form $\sigma_T(z) = za(z/T)$. We study approximations where $a(z/T)$ approaches the Heaviside function at low temperatures. Inspired by recent field theory treatments, we study the signal magnitude propagation with depth subject to small perturbations. We develop several theorems to explain their behavior and discuss how to make information propagation stable through critical initialization. We find that for small temperatures these activations remain close to ReLU in both initialization statistics and training dynamics. For temperatures of order one or higher, typical initializations (e.g., He) hinder information propagation through exponential decay of signal magnitude. On the other hand, the critical initialization stabilizes the network statistics for all temperatures and results in training curves similar to ReLU. Despite the activations never strictly becoming ReLU, they show ReLU-like properties in practice, while retaining differentiability. This is further confirmed by our neural tangent kernel analysis. At initialization, these activations display NTK values very close to ReLU, with $\tilde{\Theta}/n$ approximately constant. During training they retain a well-defined (continuous) NTK update which helps explain learning and convergence. In other words, smooth-ReLU networks have continuous updates during training, while ReLU networks have noisy updates. In future work, we would like to extend our analysis to convolutional and residual networks. It may be of interest to apply this analysis to neural ODE's, where differentiable activations play a major role. In addition, it would be interesting to analyse data normalization in conjunction with critical initialization.

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
