# OpenReview forum: "When are smooth-ReLUs ReLU-like?"
_ICLR.cc/2023/Conference — Submitted to ICLR 2023_

### Official Review · Reviewer_Tjrr · 2022-10-22

**Confidence:** 2
**Clarity, Quality, Novelty And Reproducibility:** I suspect that the paper is not suffi…
**Correctness:** 3
**Technical Novelty And Significance:** 2
**Empirical Novelty And Significance:** 2
**Recommendation:** 5

**Strength And Weaknesses:**

The subject of understanding which initialization schemes work well for smooth relaxations of ReLU is well-motivated and interesting.

I read the paper, but unfortunately, I didn’t understand it (starting from Section 3). There are two possible reasons:

(1) It is my fault, namely, I am not sufficiently familiar with this subject and/or did not devote enough time to understand the paper.

(2) The paper is not well-written.

**Summary Of The Paper:**

The paper considers several smooth relaxations of ReLU. The smoothness of these networks is controlled by a parameter called “temperature”. The authors theoretically analyze fully-connected networks with these activations and derive temperature-dependent critical initialization schemes. Namely, initializations such that the values in different layers in a deep network do not explode or vanish. The authors also demonstrate empirically that representation variance and NTK of smooth ReLUs resemble those of ReLU under the stable initialization scheme we provide, for any temperature, and show that smooth-ReLUs typically have stable and continuous NTK updates during training, while ReLU shows stochasticity instead.

**Summary Of The Review:**

I recommend weak-reject with low confidence.

---

> ### Author Response · Authors · 2022-11-17
> **Reply to Reviewer 3**
>
> We are very sorry the reviewer could not understand our paper, particularly beyond section 2 where our most interesting results lie.
> For instance, figure 1 shows a remarkable agreement between theory and experiment at initialization, figure 2 demonstrates experimentally the impact of the temperature hyper-parameter and the need for critical initialization. Moreover, figure 4 experimentally illustrates the impact that initialization has on training etc.
> Based on the suggestions of all the referees we have expanded and better explained every step we take in the paper. We urge the reviewer to read both the new version of the article and particularly the appendix for detailed explanations and derivations. In addition, our replies to the other reviewers may provide further clarification. Indeed, this work is theoretically inclined, and the material may require some time to absorb, but we have striven to clarify each point we make, and to balance theory with experiments, within our computational resources.

---

### Official Review · Reviewer_AkRp · 2022-10-25

**Confidence:** 2
**Correctness:** 3
**Technical Novelty And Significance:** 3
**Empirical Novelty And Significance:** 2
**Recommendation:** 5

**Clarity, Quality, Novelty And Reproducibility:**

Clarity: A lot to be desired. See weakneeses above for details.

Quality: Difficult to judge given many clarity issues. Of the little I could verify, I didn't find any mistakes.

Novelty: The results and analysis build on prior works (NTK and NN as Gaussian processes), but the specific application on smooth ReLUs is novel and could have significant impact on practical usage.

Reproducibility: Again, clarity issues make this not very reproducible.

**Strength And Weaknesses:**

Strengths:
* With an evergrowing zoo of activation functions, often very similar to each other, there is a growing need for a crisp theory to understand their inherent properties, and when one should prefer a specific one. The family of ReLU-like activation functions is vast, and this paper proposes a method to study a large set of them if they can be represented as $f(z, T) = z \cdot a(\frac{z}{T})$ for some sigmoid function $a(z)$ that converges to the Heaviside function for $T \to 0$. Such a general result could have a major impact on how activation functions are used and how new ones could be designed.
* The paper also examines 3 novel kinds of smooth ReLUs based on this generic definition, and examines their properties. This demonstrates the breadth of the theoretical framework.
* As a side note, I suggest the authors to consider adding to their table the Softplus function (the "original" smooth alternative to ReLU). It too can be put into the studied form by rewriting it as $Softplus(z) = z \cdot a(z/T)$ for $a(x) = \frac{\ln(1+\exp(x))}{x}$, and it can be shown that $a(x)$ converges to the Heaviside function as the temperature goes to zero, but $a(x)$ is not a sigmoid function. It is unclear in your analysis if $a(x)$ being a sigmoid is critical to your results or not, and having a function that does not follow this pattern could shed more light.

Weaknesses:
* The paper has serious clarity issues, starting with not giving a clear definition of the core expressions in its analysis. For example, a very loose definition of $< \cdot >$ is given in the text, and the definition for $< \cdot >_K$ is completely absent. One has to read one of the cited works [1] to understand what it means, and the paper doesn't even bother directing the reader to do so. Moreover, after examining the other paper that defines these terms, it raised even more questions. This is because only at the end of section 2.1 is it apparent that the prior definitions were for the limit of infinite width/depth. A much more in-depth background is needed to explain how this limit is taken. Similarly, NTK is poorly introduced, giving no insight into its definitions and why the reader should care. This paper essentially requires you to be well versed in the literature on NTK and the works on the limit of infinite NN as Gaussian processes.
* Given this point, I had a hard time understanding and verifying the correctness of this work. Therefore, I cannot judge whether the analysis is accurate or evaluate its limitations.
* Furthermore, due to the various clarity issues, many of the results are essentially left unexplained and unmotivated. This is too bad, as beyond the point on criticality, which I understood, I cannot really say I understood the point of the later sections that are based on NTK, which is a shame as I'd like to believe there might be interesting insights there.
* Of a minor note, it appears that the style file was modified. I'm not taking this into account in my rating, but simply pointing this out for the AC for their consideration.

[1] - B. Hanin. Correlation functions in random fully connected neural networks at finite width. arXiv preprint arXiv:2204.01058, 2022.

**Summary Of The Paper:**

The paper compares the ReLU activation function to its many smooth variants that converge to it in the limit of low temperatures. The paper proposes to investigate the role of these two sub-classes of activation functions by looking at their gradient propagation at initialization time. In particular, prior work has shown that there are some initializations that produce the known phenomenon of exploding/vanishing gradients. In addition, there are specific initializations where the magnitude of gradients stays constant on average throughout the network. When the latter is possible the paper defines it as a criticality. In this work this property is studied for a number of variants of smoothed ReLUs and how it changes with temperature. The paper proves the sufficient and necessary conditions for a smoothed ReLU kind of function to have a critical initialization scheme, and then examines how it manifests for the various activations, including whether this property is stable or not (stable for small pertubations). The paper continues with examining the various activation function through the lens of Neural Tanget Kernel, and use that to argue when smooth ReLU follows similar properties as the non-smooth ReLU. Finally, the paper examines how these activation functions affect the training dynamics, showing a very different behavior of the smooth vs. the non-smooth case.

**Summary Of The Review:**

Due to the severe clarity issues, I cannot recommend acceptance. It could be that a reviewer that is more versed in the prior works this work builds upon could evaluate the merits of this submission independently of its presentation, but I am (sadly) unable to do so. I hope the authors will take this criticism and revise their manuscript to be accessible to a broader readership, as what I did understand I liked.

---

> ### Author Response · Authors · 2022-11-17
> **Reply to Reviewer 2**
>
> We thank the reviewer for the positive feedback, particularly for the comment regarding the overarching goal of the paper and the impact
> that such line of work could have on the community. We reply to the comments made the reviewer in order below:
>
> 1)
> Originally, we had refrained from including  auxiliary functions a(z) that are not defined in the entire domain. For example a(z) for Softplus is not defined at z=0.
> Based on the suggestion from the reviewer, we have included Sofplus, Elu, and other activation functions in our analysis.
>
> 2)
> We are sorry to hear that the reviewer found the presentation of the manuscript unclear. Indeed, the later parts of the manuscript contain some of the most interesting results. In line with this comment and similar comments by the other reviewers, we have endeavoured
> to clarify the manuscript by properly explaining the term `ensemble average', the Guassian integration with the subscript K, and by more clearly explaining the setup both in the main text and in the appendix. Some of this was already included in the previous version in the appendix, and we thank the reviewer for pointing out these issues as now we believe the work to be self-contained and easier for the reader to follow along. Reference [1], and other references we provide in our presentation, are indeed useful for further information and more details, but should not be necessary to understand this work.
>
> 3)
> Based on the feedback of all the reviewers regarding the NTK section in the article, we have expanded and better connected it to the rest of the work.
> Regarding the comment on infinite width/depth, as mentioned in the reply to the first reviewer, although the true optimal initialization schemes will differ from those derived based on the infinite-width approximation, the corrections will generally be of order O(l/n), where l is the depth of the network and n is its width. As such, our derivations are accurate to leading order in l/n, and they are meant to be applied to finite width and depth networks with l/n smaller than 1. For such networks, the critical initialization schemes, as derived in our work, will be very good approximations, as can be seen in our initialization statistics plots. Indeed the theoretical predictions are matched quite well by the experimental results. We have made sure to further clarify this point in the respective section. We hope the reviewer will find the new version much easier to read and follow along. We have provided the step by step derivations in the appendix for clarity and hope the reviewer finds them useful. We have further expanded the explanation.
> Since it is our core belief that science should be open and reproducible, we have provided all the necessary scripts to reproduce both the results in the table and all the plots in the main text and appendices.
> We are very sorry to read that the reviewer believes the style was changed, and if so, are not unaware of how this happened. However, we have made sure the new version follows the style of the conference, and would like to point out that both previous and current versions of the article are within the allowed page number limits when compiled with the conference style.

---

### Official Review · Reviewer_S8pa · 2022-11-03

**Confidence:** 3
**Correctness:** 4
**Technical Novelty And Significance:** 3
**Empirical Novelty And Significance:** 2
**Recommendation:** 5

**Clarity, Quality, Novelty And Reproducibility:**

Clarity: The figures in the paper could have been made clearer to convey the main takeaways
Novelty: The problem of critical initializations for differentiable ReLUs has not been addressed in prior work to my knowledge

**Strength And Weaknesses:**

Strengths:
1. The paper derives critical initializations for many commonly used relaxations of ReLUs like SWISH and GeLU. The derivation seems correct.
2. The proposed initializations train better than He et. al. initialization at all temperatures in the experiments with MLPs.

Weaknesses:
1. Experiments: It would have been useful to see how these initializations perform for realistic tasks compared to standard initializations used in practice. Activations like Swish are being used for transformers widely and it would be great if a theoretically-motivated initialization can provide improved training dynamics. I understand that residual connections and normalizations add further complexities to the derivation, but it could be useful to see if the initialization matters in the practical settings anyway.
2. While it is interesting that the NTK updates look smoother for these neural networks, could the authors comment on why we should care about this quantity for practical settings? Is there an analogous quantity outside the NTK regime that could be useful to look at?
3. I find the impact of this paper to be incremental. Theoretical impact: While it is nice that these activation functions are smooth and have nicer NTK properties, I am not sure if these highlighted properties are the theoretical reason these activation functions are preferable to ReLUs in practice. Practical impact: the proposed initializations could potentially lead to faster training dynamics to He et. al., but that has not been demonstrated in this paper i.e. they show that He et. al. behaves differently for different temperatures, but the overal dynamics are not faster than He et. al. The other remaining reason this paper could be impactful is if it develops new techniques in their derivations, but I am not well versed with the literature enough to know if so.

Minor comments.
1. Would help readability if you can relabel axes in figures with what the quantity indicates. For eg: In Figure 3, replace n with ‘width’


**Summary Of The Paper:**

This paper studies temperature-dependent differentiable approximations to the ReLU activation function. The paper derives the critical initialization for this approximation at all temperatures and shows empirically that these smooth activations have similar NTK to ReLU under their proposed initialization.


**Summary Of The Review:**

While it is useful to know the critical initializations of relaxations to ReLUs being used in practice, the paper does not show that these initializations are practically useful compared to standard initializations. The nicer properties of these activations under their proposed initialization are interesting, but again do not give us much theoretical or practical insight.

---

> ### Author Response · Authors · 2022-11-17
> **Reply to Reviewer 1**
>
>
> We thank the reviewer for their feedback and are glad to hear they consider the derivations to be correct and our proposed initialization of potential impact for training.
>
> 1)
> We agree that it is interesting to extend these results to Transformers etc.
> The widespread use of smooth-ReLUs in such architectures is an important reason we are interested in studying them.
> Unfortunately, as the reviewer acknowledges, it is not straightforward to extend the theory to these cases.
> Our focus is how smooth-ReLUs can be properly initialized to avoid loss of information with depth in FFN.
> Such an analysis needs to be done for convolutions, residual connections, etc., before tackling complicated settings. We expect that the critical initialization schemes will be significantly affected by architectural changes.
>
> Also, the computational cost of full-batch training renders comprehensive empirical testing intractable to us. Our experimental protocol was designed to minimize the variability of training by removing the stochasticity of gradient evaluation and restricting experiments to simple and well-understood architectures. However, if applied to more complex architectures and datasets, it would be computationally expensive and beyond the resources available to us.
>
> Following the suggestion of the reviewer, we have included empirical demonstrations with EfficientNet on CIFAR100 and Transformers on WMT'14 English to German translation using GudermanLU and SWISH instead of ReLU. We also made additional initialization experiments with GumbelLU on FFN. Both experiments confirm our expectation that activations with critical values closer to ReLU have more stable training under He, and lowering the temperature helps.
>
> 2)
> Here we understand `the NTK regime' to mean the infinite-width limit and/or the linear approximation of training dynamics near the initial state of the network at the onset of training. Indeed, this regime is known to be a poor approximation of real networks.
>
> Although the true optimal initialization schemes will differ from those derived based on this approximation, the corrections will generally be of order $\mathcal{O}(l/n)$, where $l$ is the depth and $n$ the width.
> For networks with $l/n$ much smaller than 1, our initialization should be a good approximation, as shown by our initialization plots. For the reader interested in the gap between real networks and the NTK regime throughout the course of training,
> we plot the NTK for the whole training dynamics and our NTK is iteration dependent. Analogous quantities worth studying are the differentials of the NTK and the Fisher Information Matrix.
>
> ReLU FFNs have highly irregular NTK variations over time with not well-defined differentials. Networks with smooth-ReLUs have smoothly varying NTKs during training which is crucial for a theoretical understanding. The work of Roberts et al establishes the relationships between the differentials of the NTK with respect to training iteration but can not be applied to ReLU. Training theory is applicable to smooth-ReLU FFNs at all temperatures. If we can understand the convergence of these networks to ReLU ones, we could use this to extend the training theory to ReLU networks. We believe the irregularity of ReLU-network NTK differentials corresponds precisely to the stochastic/chaotic-like patterns visualized of Figure [5] in our paper. As such, we believe our contribution is an important step towards a full theory of the training dynamics of ReLU networks.
>
> 3)
> We agree with the reviewer and suspect that the reason for the widespread use of smooth-ReLUs is their differentiability. We were motivated by establishing more effective initialization schemes for these activations, providing a systematic framework to derive stable initialization, and extend the theoretical understanding of training dynamics in ReLU or smooth-ReLU networks.
>
> We improved the clarity of the presentation to show that critical initialization indeed leads to faster training dynamics for smooth-Relu activations with respect to He. Figure 4 in our paper explicitly compares them. The reviewer correctly notes that the behaviour of the smooth-ReLU networks is different from the ReLU-based network, indicated by the dotted red line.
>
> However, the point of comparison should be made between smooth-ReLU networks trained from He initializations to the same networks at the same temperature, trained with our proposed schemes.
> At high temperatures, the networks train much faster and more consistently using our proposed scheme than He. At $T=1$, for instance, the networks reach a loss of $10^{-1}$ approximately five times faster with our method. Smooth-ReLU networks are often used with temperatures near 1 as direct ReLU substitutes, initialized based on He. Our results could lead to substantially faster and more robust training for such applications.
> We have also updated the plot labels accordingly with the reviewer suggestions.

---

### Decision · Program_Chairs · 2023-01-20

**Decision:**

Reject

**Justification For Why Not Higher Score:**

There is a clear clarity issue the authors need to work on, which was raised by two of the reviewers. Although, one reviewer with expertise in the topic could understand the main points, the paper as is required significant knowledge of related works and not applicable for a broad ICLR audience. Beyond clarity, reviewers raised issues that many results are unexplained or unmotivated. Finally, there is weakness on practicality and impact.

In the end, all reviewers rated the paper as below acceptance threshold thus not ready for publication at ICLR at this moment.


**Justification For Why Not Lower Score:**

N/A

**Metareview: Summary, Strengths And Weaknesses:**

The paper studies several smooth relaxation to ReLU activation function which is controlled by a temperature parameter. For fully connected networks, authors theoretically analyze these activations and derive temperature dependent critical initialization schemes. Criticality is based on forward and backward pass of deep networks not to explode or vanish. Authors examine these activation functions using Neural Tangent Kernel and study training dynamics between smooth and non-smooth cases.

Strength
- Theory of activation is well motivated.
- Theoretical breadth by covering 3 novel kinds of smooth ReLU and derive critical initialization for commonly used relaxation of ReLUs.
- Well motivated and interesting problem of understanding initialization of networks with smooth relaxation of ReLU
- Empirical benefit over He initialization at all temperatures in the experiments with MLPs


Weakness
- Severe clarity issues: One of the reviewers had a hard time understanding the paper and other reviewers also raised serious clarity issues. This is exacerbated by the fact that many concepts are not introduced and rely on familiarity with NNGP and NTK.
- Many results are left unexplained or unmotivated
- Issues with practicality and impact: Experiments are not done in realistic tasks used in practice and only proven useful in toy-ish settings. Theoretical justification doesn't seem to explain why smooth versions of ReLU are used in practice. Also practical impact hasn't been fully demonstrated.